# Lost in Transition: Insights from a Retrospective Chart Audit on Nutrition Care Practices for Older Australians with Malnutrition Transitioning from Hospital to Home

**DOI:** 10.3390/nu16162796

**Published:** 2024-08-22

**Authors:** Kristin Gomes, Jack Bell, Ben Desbrow, Shelley Roberts

**Affiliations:** 1School of Health Sciences and Social Work, Griffith University, Gold Coast Campus, Southport, QLD 4222, Australia; jack.bell@health.qld.gov.au (J.B.); b.desbrow@griffith.edu.au (B.D.); s.roberts@griffith.edu.au (S.R.); 2Allied Health Research Collaborative, The Prince Charles Hospital, 627 Rode Road, Chermside, QLD 4032, Australia; 3Allied Health Research, Gold Coast Hospital and Health Service, 1 Hospital Blvd, Southport, QLD 4219, Australia

**Keywords:** malnutrition, hospital to home, discharge, nutrition care, older adults, care transitions

## Abstract

Care transitions from hospital to home for older adults with malnutrition present a period of elevated risk; however, minimal data exist describing the existing practice. This study aimed to describe the transition of nutrition care processes provided to older adults in a public tertiary hospital in Australia. A retrospective chart audit conducted between July and October 2022 included older (≥65 years), malnourished adults discharged to independent living. Dietetic care practices (from inpatient to six-months post-discharge) were reported descriptively. Of 3466 consecutive admissions, 345 (10%) had a diagnosis of malnutrition documented by the dietitian and were included in the analysis. The median number of dietetic visits per admission was 2.0 (IQR 1.0–4.0). Nutrition-focused discharge plans were inconsistently developed and documented. Only 10% of patients had nutrition care recommendations documented in the electronic discharge summary. Post-discharge oral nutrition supplementation was offered to 46% and accepted by 34% of the patients, while only 23% attended a follow-up appointment with dietetics within six months of hospital discharge. Most patients who are seen by dietitians and diagnosed with malnutrition appear lost in transition from hospital to home. Ongoing work is required to explore determinants of post-discharge nutrition care in this vulnerable population.

## 1. Introduction

Older adults are at a greater risk of malnutrition secondary to age-related physiological, psychological, social, and economic changes that may negatively impact their nutritional intake [1,2,3]. Malnutrition can lead to significant subcutaneous fat loss and muscle wasting, and increased risks of infections, complications (including pressure injuries and falls), and mortality [2,3]. As the global population continues to rapidly age [4], the prevalence of chronic diseases associated with conditions such as malnutrition is likely to rise. Malnourished adults spend more time in hospital or rehabilitation, are more likely to be readmitted, require more primary care visits, and have a reduced quality of life [3,5]. The poor clinical outcomes experienced by patients and the increased healthcare costs associated with malnutrition place a significant burden on individuals, their families, and healthcare systems [1,5,6,7].

The prevalence of malnutrition amongst hospitalised older adults (aged ≥ 65) has been reported to range from 20 to 50% [3], with significant variations in rates observed across regions [8]. For instance, in Europe and North America, prevalence rates of 32–58% have been reported in hospitalised older adults [9,10,11]; in Latin America, between 40 and 60% [12]; and in northeast and southeast Asia, rates exceed 60% [13]. In Australia, malnutrition affects 30–40% of hospitalised older patients [1]. As most older adults live in the community, malnutrition often arises there first and is present upon admission to hospital [14,15,16]. The prevalence of malnutrition among community-dwelling older adults varies globally, with rates reported to range between 1% and 26% [17,18,19,20]. Further, the prevalence of malnutrition among European community-dwelling adults aged ≥75 years is thought to be two times higher than those aged 65–74 years [21]. Despite these regional variations, malnutrition in older adults is frequently overlooked in community settings and is typically only identified and managed during hospitalisation [17,22].

While a hospital admission provides an opportunity to identify and treat malnutrition, extended periods of fasting, interrupted meals, poor appetite, and a low acceptance of hospital food can exacerbate the poor nutritional status that is present on admission [3,23,24]. Moreover, as the average length of stay in most public hospitals continues to decrease, it is unlikely that malnutrition will be resolved in the acute care setting [25]. Therefore, ensuring the continuity of nutrition care post-discharge is critical to improving patient recovery and optimising their health outcomes [26,27].

Randomised controlled trials investigating the impact of various post-discharge nutrition interventions have demonstrated significant and clinically relevant increases in mean daily energy and protein intake and body weight [28,29,30], improvements in quality of life [31], and reduced long-term mortality in older adults with or at risk of malnutrition [28]. However, the continuation of nutrition care after hospital discharge depends on the development and dissemination of nutrition-focused discharge plans, effective coordination of the transition of nutrition care post-discharge, and the capability and motivation of patients to participate in this care [32,33]. Few studies have examined post-discharge nutrition care practices among older adults with or at risk of malnutrition, but it appears that these patients do not consistently receive nutrition care plans or recommendations that could improve their outcomes, and even if they do, they may not adhere to them [26,27,34,35,36]. This is concerning, given that older patients with or at risk of malnutrition often experience further weight loss within 30 days of being discharged from hospital [37]. Of the five completed studies, three focused on describing nutrition care provided to older adults transitioning to home from the hospital through targeted interventions distinct from usual clinical practice [26,34,36]. In contrast, only two aimed to describe the nutrition care delivered as part of usual clinical practice to this target demographic [27,35]. These were small, single-site studies, one of which involved adults aged ≥18 years, that focused either exclusively on examining dietary changes documented in electronic discharge summaries or investigating patient-reported follow-up with dietetics within 30 days of hospital discharge alongside changes in functional and clinical outcomes [27,35]. Furthermore, neither of these studies examined practices related to nutrition-focused discharge planning and post-discharge nutrition care coordination, or older adults’ engagement in post-discharge nutrition care [27,35]. A recent study by Bell et al. examining the engagement of malnourished inpatients in nutrition care processes revealed that 40% reported not having a plan for ongoing nutrition care and follow-up after hospital discharge [38]. It is unclear whether this issue was due to poor patient recall or a failure by the designated healthcare professionals to action the necessary care. As such, a knowledge gap persists regarding the planning, coordination, and receipt of standard nutrition care practices aimed at preventing or treating malnutrition in older adults transitioning from hospital to home.

To effectively address such challenges and identify opportunities for enhancing nutrition care to meet the needs of older adults, it is essential to first understand current practice [39]. We hypothesise that post-discharge nutrition care is suboptimal. This study aims to describe the current nutrition care provided to older adults with malnutrition transitioning from hospital to home, including inpatient nutrition care, nutrition-focused discharge planning and coordination, and the receipt and uptake of dietetic referrals post-discharge. Understanding these aspects is essential to determine if future interventions are needed to enhance staff capacity and care quality. The findings of this study will guide efforts to enhance post-discharge nutrition care for older adults with malnutrition.

## 2. Materials and Methods

### 2.1. Study Design

A retrospective chart audit was undertaken. Ethical approval was provided by Gold Coast Hospital and Health Service and Griffith University Human Research Ethics Committees.

### 2.2. Setting

This study was conducted in a 900-bed public tertiary hospital in [blinded], [blinded]. The Nutrition and Dietetics department employs 21.2 full-time-equivalent (FTE) dietitians who oversee nutrition care across twenty acute adult medical or surgical units, excluding maternity and mental health. Additionally, the department manages seven outpatient specialist dietetic clinics and conducts research in the inpatient and outpatient/community settings. Adult inpatients from the local catchment area can also be referred to one of three community dietetic clinics within the health service.

### 2.3. Participants

Adult inpatients aged ≥ 65 years admitted to a medical or surgical ward at [blinded for peer review] between 1 July 2022 and 1 October 2022 were included. Eligibility was determined using the following criteria:

Inclusion Criteria

(1)Admitted from home and discharged home to independent living in the community.(2)Admitted to acute medical or surgical wards.(3)Seen by a dietitian during admission.(4)Had a documented diagnosis of malnutrition by the dietitian in a chart entry for that admission.

Exclusion Criteria

(1)Admitted to psychiatric or intensive care units.(2)Admitted for day surgery or day procedures.(3)Discharged home with ‘Hospital in the Home (HITH)’ or similar services.(4)If their place of residence subsequently changed to a residential aged care facility prior to discharge.

These inclusion/exclusion criteria were intended to identify older adults capable of continuing their nutrition care upon transitioning home with appropriate recommendations and follow-up as required.

### 2.4. Data Collection

De-identified patient data were extracted in two phases from the hospital’s integrated electronic medical record (ieMR) system. In phase I, de-identified patient data were extracted from the ieMR into a structured dataset and provided to the research team. The dataset included patient demographics (age, sex, marital status, discharge destination), admission information (length of stay, number of admissions within study timeframe), and inpatient nutrition care data (malnutrition diagnosis, dietetic input from a dietitian or dietitian assistant during admission). One researcher, an accredited practicing dietitian with inpatient clinical experience, reviewed the dataset. First, any incomplete data were flagged, and ineligible admissions were excluded. Next, admissions not seen by dietetics were excluded. For the remaining admissions, if malnutrition diagnosis data were absent from the initial dataset, the researcher addressed this by manually reviewing dietitian chart entries in the ieMR to verify the presence of the diagnosis in these entries and document this in the dataset where available. Within the facility, dietitians use the Subjective Global Assessment (SGA) tool to assess and categorise patients as A (well nourished), B (mild/moderately malnourished), or C (severely malnourished) [40].

In phase II, admissions were excluded if they were assessed as well-nourished or lacked a documented malnutrition diagnosis by the dietitian for the admission. The final study population included only those patients diagnosed and documented as malnourished by the dietitian. For all admissions included in the final study population, the researcher manually reviewed the ieMR and extracted free-text nutrition care data from hospital dietitians’ chart entries. These data encompassed nutrition care delivered during the admission, nutrition-focused discharge planning, and post-discharge dietetic follow-up, ensuring a comprehensive dataset for analysis. To ensure all documentation relating to nutrition-focused discharge planning and coordination of post-discharge nutrition care was captured, the researcher explored all relevant components within the ieMR platform including hospital dietitian chart entries, outpatient and community dietitian progress notes (where available), and the ‘Patient Appointments’ and ‘Enterprise Discharge Summary’ (EDS) tabs.

Within the ieMR, the EDS is available for interdisciplinary treating staff to complete. It is mandatory for all patients admitted to [blinded] to receive an electronic discharge summary, regardless of their length of stay, discharge destination, or discharge status [41]. The EDS was designed to create and securely transfer discharge summaries to GPs, serving as the primary source of clinical communication with the GP [41]. It summarises inpatient care and post-discharge recommendations for both GPs and patients [41]. The EDS associated with each admission was reviewed to identify any nutrition-focused discharge planning or recommendations. A detailed summary of the data extracted in phases I and II is included in Appendix A.

### 2.5. Data Analysis

The data were entered into SPSS Statistics for Windows version 23.0 (IBM Corp. 2012, Armonk, NY, USA) for analysis. The normality of the data was verified using the Shapiro–Wilk test. Descriptive statistics were used to report the characteristics of the study population, the nutrition care provided during admission, nutrition-focused discharge planning, and dietetic follow-up. Continuous variables were presented as mean ± standard deviation (SD) (or median and interquartile range (IQR) for non-normally distributed data) and frequency (%) for categorical data.

## 3. Results

Of the 3466 eligible admissions, 345 (10%) included a documented diagnosis of malnutrition by a dietitian and were included in the final analysis. Figure 1 summarises the sample selection process for phases I and II.

### 3.1. Participant Characteristics

The characteristics of the study population are summarised in Table 1. The mean age was 77.0 ± 7.4 years (range 65–94 years) and the median length of stay was 9.0 (IQR 5.0–16.0) days. The majority were male, were married, had a single admission during the study period, and were discharged directly home.

Of the 306 individual patients who made up the 345 admissions, 105 (34%) were deceased within six months of discharge from their final hospital admission. The median time from hospital discharge to death was 21 (IQR 1.0–86.0) days.

### 3.2. Inpatient Nutrition Care

Dietitians and dietitian assistants completed a total of 1183 visits. The median number of dietetic visits per admission was 2.0 (IQR 1.0–4.0). Dietitians conducted 83% of the visits, while dietitian assistants conducted 17% of the visits; 73% of all visits were face-to-face with patients. Over 80% of patients received two or more strategies for food and nutrient delivery, including oral nutrition supplements (ONS) (77%), high-protein high-energy meals and extras (67%), or enteral/parenteral nutrition (9%). Just over half of the patients (60%) had a documented chart entry indicating they had received verbal education regarding the importance of maintaining their protein and energy intake during their admission to support recovery and prevent further weight loss.

### 3.3. Post-Discharge Nutrition Care and Follow-Up

#### 3.3.1. Nutrition-Focused Discharge Planning

Dietitians lacked a standardised approach to documenting nutrition-focused discharge plans, leading to inconsistencies in the format, process, and location of these within the ieMR. Recommendations for post-discharge nutrition care and follow-up were documented in the chart entry at different stages: during the initial dietitian assessment, at a subsequent or final dietitian review, not at all, or exclusively in the EDS prior to discharge. When recorded in the dietitian’s chart entry, these recommendations appeared under headings such as ‘Diet’, ‘Assessment’, ‘Intervention/Plan’, or ‘Monitoring/Follow-Up’, often intermingled with recommendations for ongoing inpatient nutrition care. This wide variation in documentation practices posed challenges in identifying and assessing the presence, frequency, and quality of nutrition-focused discharge plans documented in the chart notes. Upon reviewing the EDS for each of the 345 admissions with documented malnutrition, only 35 (10%) had post-discharge nutrition care recommendations documented by a dietitian.

#### 3.3.2. Food First Approach after Hospital Discharge

Prior to discharge, 116 (34%) admissions with documented malnutrition were provided with printed nutrition education materials as documented in a chart entry by a dietitian. These materials contained recommendations for adopting a high-protein high-energy diet at home to support weight gain, build muscle strength, and aid in recovery from acute illness. The strategies outlined in these resources focused on eating small and frequent high-protein and -energy meals and snacks, fortifying foods and beverages, prioritising protein intake during meals, and choosing nutrient-dense fluids such as flavoured milk and juice over tea, coffee, or water. Additionally, for 22 (6%) admissions, dietitians documented a recommendation to access community services such as transportation, shopping assistance, and/or home meal delivery.

#### 3.3.3. Oral Nutrition Supplementation after Hospital Discharge

Of the 345 admissions with documented malnutrition, 157 (46%) were offered a prescription for ONS after discharge from hospital. Of these, 116 (74% of those offered) accepted the prescription. There was significant variation regarding the type and dose of ONS prescribed by the dietitian. Overall, 34% of admissions with documented malnutrition accepted an ONS prescription after discharge from hospital.

#### 3.3.4. Dietetic Follow-Up after Hospital Discharge

Figure 2 shows the proportion of admissions with documented malnutrition who were offered, accepted, and attended a follow-up appointment with community or outpatient dietetics within six months of hospital discharge. The timeframe for follow-up after discharge varied between one day and six months. Of the 345 admissions with documented malnutrition, 37 (11%) were placed in short-term specialised support programs for older adults upon discharge from the hospital, which included access to dietetic services in the home with patient consent. In total, 79 (23%) admissions with documented malnutrition attended a follow-up appointment with dietetics. Most appointments occurred within one month of hospital discharge, primarily in specialist outpatient dietetic clinics, with follow-up ranging from one day to six months post-discharge.

## 4. Discussion

This study explored the nutrition care provided in a large public tertiary hospital to older adults with malnutrition transitioning from hospital to home, including inpatient nutrition care, nutrition-focused discharge planning, post-discharge nutrition care coordination utilising ieMR systems, and patient receipt and uptake of dietetic referrals post-discharge. The results showed that most patients received a nutrition care plan, nutrition interventions such as food fortification and ONS, and verbal nutrition education during their hospital stay. In contrast, nutrition-focused discharge planning and documentation of post-discharge nutrition care recommendations varied widely. Few patients received nutrition-focused discharge plans, post-discharge dietary counselling, guidance on accessing community supports, prescriptions for ongoing ONS use, or follow-up with dietetics.

To our knowledge, this is the first study to retrospectively examine and compare inpatient and post-discharge nutrition care practices delivered to older adults with malnutrition transitioning from hospital to home. This study found that 6–34% of patients received post-discharge nutrition recommendations specific to malnutrition, 34% accepted a prescription for ONS, and only 23% attended a follow-up appointment with a dietitian after hospital discharge. In this respect, our findings are comparable to the few studies that have examined nutrition care practices during the hospital-to-home transition [26,27,34,35,36]. Former studies report that less than 50% of older patients with or at risk of malnutrition received discharge plans or recommendations specific to malnutrition [26,27,34,38], fewer than 20% had access to nutrition supports such as home meal delivery or shopping assistance after discharge [35], and only 11–14% consulted a dietitian [34,35]. This may explain why older adults with malnutrition often lose additional weight within the initial month of hospital discharge [37]. However, this study also reveals novel insights into nutrition-focused discharge planning and care coordination practices. These include the absence of a standardised approach to documenting and disseminating nutrition-focused discharge plans and limited use of the EDS to communicate nutrition care to primary care physicians or specialists post-discharge. Additionally, it highlights discrepancies between the inpatient dietitian care provided and the provision and uptake of referrals for outpatient dietetic follow-up, along with reasons for non-attendance. These observations highlight areas not previously explored in the literature. Finally, this study reveals a concerning mortality rate (34%) among these patients within six months of discharge. Collectively, it appears that current nutrition care practices delivered to older adults with malnutrition as they transition from hospital to home is suboptimal, as hypothesised, and requires improvement.

The continuation of nutrition care after hospital discharge is contingent upon several factors. These include (1) the development and dissemination of nutrition-focused discharge plans by hospital clinicians (e.g., dietitians), (2) effective interdisciplinary communication and collaboration, and (3) the capability and motivation of patients to participate in post-discharge nutrition care [32,33]. The findings from the current study suggest that challenges exist with all of these factors, and efforts to improve care will require an understanding of barriers associated with each. For example, the variation in and absence of a standard approach to documenting nutrition-focused discharge plans in the ieMR system, as well as underutilisation of the EDS, may contribute to identified barriers such as a lack of standardised discharge planning policies and procedures [42], poor interdisciplinary communication [42,43], insufficient time [44,45], and/or a lack of standardised nutrition care pathways to ensuring the continuity of nutrition care for older adults transitioning from hospital to home [32]. Addressing and overcoming barriers associated with the dissemination and communication of information between health professionals and understanding patient capability and motivation are essential for enhancing discharge nutrition care practices.

Clinical nutrition research over the past few decades has focused on assessing nutrition care practices in the hospital setting [8]. This research has helped shape standards set by clinical governance bodies responsible for overseeing the quality and delivery of hospital care. Indeed, clinical practice guidelines [46,47,48], best practice resources [49], and models of care [50,51,52] exist to guide healthcare professionals to deliver quality nutrition care to malnourished older patients in the hospital setting. The results from the current study indicate inpatient nutrition care practices align with those demonstrating improved patient outcomes for hospitalised older adults with or at risk of malnutrition [53,54,55]. In contrast, limited research on optimising post-discharge nutrition care practices for older adults with malnutrition transitioning home from hospital has been undertaken [26,36]. Consequently, current clinical nutrition guidelines lack specific recommendations for discharge planning in older adults with or at risk of malnutrition [46,47,48]. Most guidelines are limited to general statements acknowledging the importance of commencing and sustaining nutrition care initiated during the hospital stay upon discharge to the home environment [46,47,48]. Without evidence, ambiguity will remain regarding how clinicians should approach discharge planning, determine and coordinate ongoing nutrition care, and identify the appropriate community healthcare providers to be involved in managing the transition of care.

Hospital settings have traditionally been the focal point for most research and practice improvement initiatives for managing malnutrition. However, the period immediately following hospitalisation may be a more favourable time for effective malnutrition interventions for older adults. While hospitalisation allows for the identification and initial treatment of the condition, older adults frequently experience suboptimal nutritional intake and a worsening of their nutritional status during their brief hospital stays [23,37]. Patients manage competing priorities while acutely ill [56], with malnutrition typically not the primary reason for their hospitalisation [57], thus making nutrition care a low patient priority in this setting [58,59]. Moreover, as the average length of stay in most public hospitals continues to decrease, malnutrition will not likely be resolved in the acute setting [25]. Consequently, older adults often face diminished physical reserves and ongoing disease-related symptoms upon discharge, making the transition from hospital to home a critical time for these patients [23,24,60].

Following hospital discharge, older adults may be more receptive to nutrition care, as they return to a familiar environment and are motivated to resume their normal daily activities [61]. Post-discharge nutrition care interventions may also be tailored to consider the individual’s home environment, unique dietary preferences, and ability to manage their nutritional needs independently or with support [62]. Early post-discharge nutrition care is crucial, given the potential for further weight loss [37] and challenges like disrupted care continuity, limited access to community services, and financial constraints [23,24,60]. Prioritising improvements in post-discharge nutrition care can help to mitigate these issues, significantly enhancing patients’ recovery, optimising their health outcomes, and supporting them to regain independence and resume their normal daily activities sooner. Nevertheless, this study highlights a notable decline in older adults’ engagement with nutrition care after hospital discharge. Therefore, understanding the priorities, goals, and key barriers to sustaining nutrition care post-discharge for this patient population is critical.

## 5. Strengths and Limitations

The findings of this study reveal gaps in nutrition-focused discharge planning, post-discharge nutrition care coordination, and the receipt and uptake of dietetic referrals among older adults with malnutrition transitioning from hospital to home. The results highlight opportunities for future research to enhance post-discharge nutrition care for older adults with malnutrition. However, this study has some limitations. The exclusion of patients not seen by dietitians or without a documented nutrition diagnosis, accounting for three-quarters of the admissions during this period, may have led to an overestimation of the provision of post-discharge nutrition care. This study was also conducted at a single site. Clinical contexts vary widely across healthcare settings, impacting healthcare professionals’ approaches to clinical practice, thus limiting the generalisability of the findings to other clinical settings and locations. Further, one member of the research team manually extracted free-text nutrition care data from the ieMR system, involving subjective interpretation of varied documentation quality among clinical dietitians. This variability posed challenges for objectively assessing the extent of nutrition-focused discharge planning and coordination, particularly in cases with vague descriptions, potentially leading to under- or overestimations of the nutrition-focused discharge planning and post-discharge nutrition care provided.

## 6. Conclusions

In conclusion, we know nutrition care for community-dwelling older adults with malnutrition is lost in transition as they move from hospital to home. The next step is to conduct research designed to better support staff to ensure effective nutrition care continues from the hospitals to older adults’ homes.

## Figures and Tables

**Figure 1 nutrients-16-02796-f001:**
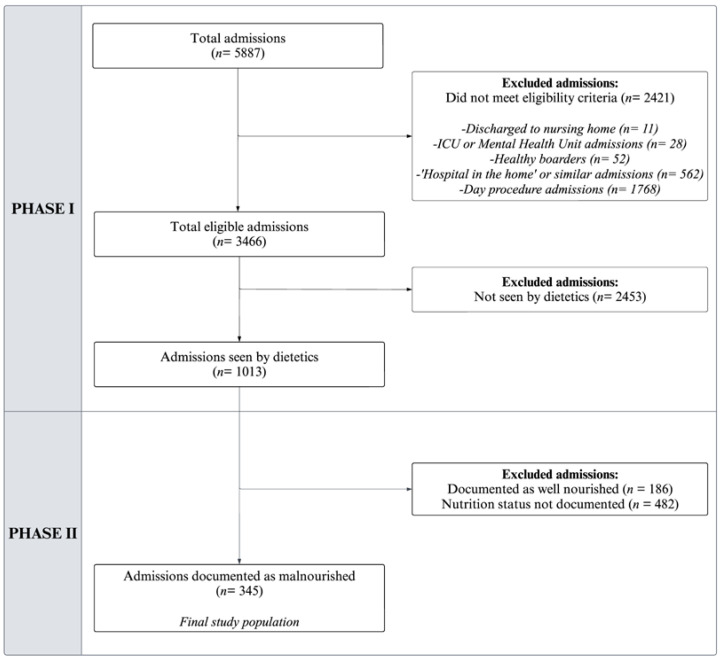
Consort flow diagram for selection of final study population from 5887 admissions of adults aged ≥65 years, admitted between 1 July and 1 October 2022.

**Figure 2 nutrients-16-02796-f002:**
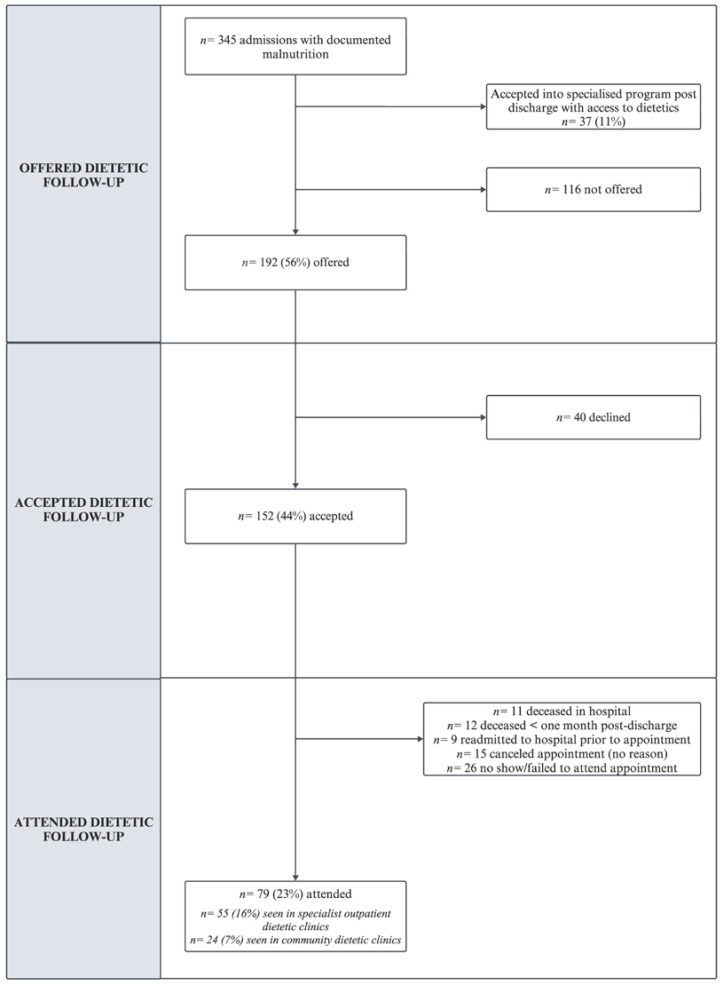
Proportion of admissions with documented malnutrition who were offered, accepted, and attended dietetic follow-up in community or outpatient setting within six months of hospital discharge.

**Table 1 nutrients-16-02796-t001:** Participant demographics.

Admissions with Documented Malnutrition (*n* = 345)		*n* (%)
Sex	Male	188 (54)
	Female	157 (46)
Marital status	Married/Defacto	167 (49)
	Separated/Divorced/Never Married	118 (34)
	Widowed	53 (15)
	Not stated	7 (2)
Total admissions per patient	Patients with a single admission	272 (89) ^a^
	Patients with two admissions	29 (9) ^a^
	Patients with three admissions	5 (2) ^a^
Discharge destination	Home	291 (84)
	Died in hospital	34 (10)
	Hospital transfer	20 (6)

^a^ of 306 individual patients.

## Data Availability

The study data are available upon request from the corresponding author, KG. The data are not publicly available due to the sensitive nature of the information (i.e., patient-related data). This decision aligns with the privacy and confidentiality policies mandated by the Ethics Committees that approved this study.

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
