# Peer review of "Lost in Transition: Insights from a Retrospective Chart Audit on Nutrition Care Practices for Older Australians with Malnutrition Transitioning from Hospital to Home"

_nutrients, 2024, doi:10.3390/nu16162796_

Round 1

Reviewer 1 Report

Comments and Suggestions for Authors

The authors evaluated malnutrition Transitioning from Hospital to Home by presenting a retrospective chart audit on nutrition care practices for 3,466 hospitalized older adults. The quality of life and day-to-day activities of older persons are negatively impacted by malnutrition. The paper is very interesting and treat a very important topic. Screening for malnutrition in hospitalized geriatric populations is essential for developing appropriate interventions and care plans, as these people are more vulnerable to malnutrition because of age-related variables and complex diseases (10.1016/s0140-6736(22)02612-5). The importance of an appropriate nutrition in continuity of care is crucial, and directly impact on quality of life. Unfortunately, frail patients often do not follow physicians suggestions and a continue education is very important.

Author Response

Comment 1: 

Reviewer 1 - Comment 1:

The authors evaluated malnutrition Transitioning from Hospital to Home by presenting a retrospective chart audit on nutrition care practices for 3,466 hospitalized older adults. The quality of life and day-to-day activities of older persons are negatively impacted by malnutrition. The paper is very interesting and treat a very important topic. Screening for malnutrition in hospitalized geriatric populations is essential for developing appropriate interventions and care plans, as these people are more vulnerable to malnutrition because of age-related variables and complex diseases (10.1016/s0140-6736(22)02612-5). The importance of an appropriate nutrition in continuity of care is crucial, and directly impact on quality of life. Unfortunately, frail patients often do not follow physicians suggestions and a continue education is very important.

Authors' Response

Reviewer 2 Report

Comments and Suggestions for Authors

Dear authors, thank you for submitting the manuscript "Lost in transition: Insights from a retrospective chart audit on nutrition care practices for community-dwelling older adults with malnutrition transitioning from hospital to home". Overall, I enjoyed reading your study and here is my feedback:

-The title should be shorter and indicate the location of the study because your findings may not apply worldwide.

-Abstract needs to mention the country or region of the country where the study was performed.

-Introduction has many facts without being specific of the location, specially the second paragraph, I may guess that those percentages are not the same in different regions, example North America to South America and to Africa. Provide values from different regions and it can lead you a better worldwide values.

-I like how you ended up your introduction with all the information related to the randomised trials and other studies. Please include your hypotheses at the end.

-For the materials and methods, create a small table describing your inclusion and exclusion criteria for the patients, it will make it easier for the reader.

-The flowchart should be more like a Graphic Abstract (GA) in which you can describe the workflow of the study. The GA should be inserted in the results and it should include the setting, number of participants, data analysis, and results. GA aims to give a quick and great understanding of the manuscript without the need of reading the 14 pages.

-I like how you described your strengths, limitations and further studies you would like to aim in the future.

-Make conclusion shorter, just a a couple of sentences, not an entire paragraph.

Author Response

Comment 1:

Dear authors, thank you for submitting the manuscript "Lost in transition: Insights from a retrospective chart audit on nutrition care practices for community-dwelling older adults with malnutrition transitioning from hospital to home". Overall, I enjoyed reading your study and here is my feedback

Authors' Response:

Comment 2:

The title should be shorter and indicate the location of the study because your findings may not apply worldwide.

Authors' Response:

Comment 3:

Abstract needs to mention the country or region of the country where the study was performed.

Authors' Response:

Comment 6:

For the materials and methods, create a small table describing your inclusion and exclusion criteria for the patients, it will make it easier for the reader.

Authors' Response:

Comment 7:

The flowchart should be more like a Graphic Abstract (GA) in which you can describe the workflow of the study. The GA should be inserted in the results and it should include the setting, number of participants, data analysis, and results. GA aims to give a quick and great understanding of the manuscript without the need of reading the 14 pages.

Authors' Response:

Comment 8:

I like how you described your strengths, limitations and further studies you would like to aim in the future.

Authors' Response:

Comment 9:

Make conclusion shorter, just  a couple of sentences, not an entire paragraph.

Authors' Response:
